# Patients’ Post-/Long-COVID Symptoms, Vaccination and Functional Status—Findings from a State-Wide Online Screening Study

**DOI:** 10.3390/vaccines11030691

**Published:** 2023-03-17

**Authors:** Sonia Lippke, Robin Rinn, Christina Derksen, Alina Dahmen

**Affiliations:** 1School of Business, Social & Decision Sciences, Constructor University Bremen, 28759 Bremen, Germany; 2Klinikum Wolfsburg, 38440 Wolfsburg, Germany

**Keywords:** SARS-CoV-2, breakthrough infections, perceived symptom severity, social participation, workability, life satisfaction, timepoint, number of doses, age, sex

## Abstract

(1) Background: Better understanding of post-/long-COVID and limitations in daily life due to the symptoms as well as the preventive potential of vaccinations is required. It is unclear whether the number of doses and timepoint interrelate with the trajectory of post-/long-COVID. Accordingly, we examined how many patients positively screened with post-/long-COVID were vaccinated and whether the vaccination status and the timepoint of vaccination in relation to the acute infection were related to post-/long-COVID symptom severity and patients’ functional status (i.e., perceived symptom severity, social participation, workability, and life satisfaction) over time. (2) Methods: 235 patients suffering from post-/long-COVID were recruited into an online survey in Bavaria, Germany, and assessed at baseline (T1), after approximately three weeks (T2), and approximately four weeks (T3). (3) Results: 3.5% were not vaccinated, 2.3% were vaccinated once, 20% twice, and 53.3% three times. Overall, 20.9% did not indicate their vaccination status. The timepoint of vaccination was related to symptom severity at T1, and symptoms decreased significantly over time. Being vaccinated more often was associated with lower life satisfaction and workability at T2. (4) Conclusions: This study provides evidence to get vaccinated against SARS-CoV-2, as it has shown that symptom severity was lower in those patients who were vaccinated prior to the infection compared to those getting infected prior to or at the same time of the vaccination. However, the finding that being vaccinated against SARS-CoV-2 more often correlated with lower life satisfaction and workability requires more attention. There is still an urgent necessity for appropriate treatment for overcoming long-/post-COVID symptoms efficiently. Vaccination can be part of prevention measures, and there is still a need for a communication strategy providing objective information about the usefulness and risks of vaccinations.

## 1. Introduction

Post-/long-COVID is a collective term referring to different physical and psychological symptoms following a former acute SARS-CoV-2 infection [1]. Long-COVID can be attributed to symptoms remaining more than four weeks after the acute infection. In contrast, post-COVID refers to symptoms that develop within two months after the infection and last at least two months [2]. The most common post-/long-COVID symptoms include neurological symptoms, such as fatigue, memory, or concentration problems; respiratory symptoms, such as shortness of breath; or physical symptoms, such as muscle pain [2,3,4,5]. Additionally, the symptoms are not only diverse in their expression but also their severity. Patients perceive symptoms differently, with varying trajectories over time [6,7].

Patients who experience severe post-/long-COVID symptoms are affected in their functional status, including impairments to their ability to engage in daily activities and participate in social life and work [3,8,9,10]. Severe symptoms correlate with patients’ health and functional status [7]. Furthermore, it was noted that patients perceive their neurological or cognitive symptoms as debilitating or embarrassing, so the respective symptoms keep them from being part of social activities and their ability to work [11]. 

Similarly, research on life satisfaction found a bidirectional association between clinical and psychological factors, indicating that post-/long-COVID symptoms affect mental well-being but also the other way around, illustrating a vicious circle [12] in terms of that experiences feed back into setting goals towards adopting and maintaining behaviors [13]. This is especially important in patients who suffer from post-/long-COVID fatigue, associated with subsequent mental health issues, and if patients do not recover well (e.g., [3,14]). 

Consequently, post-/long-COVID symptoms can lead to disability and a long-term burden for the individual and society. This is particularly significant for the healthcare system, as previous research has determined that approximately 10–20% of all infected individuals suffer from post-/long-COVID symptoms and require support [6,15,16,17,18]. Therefore, studies are needed that examine post-/long-COVID symptoms and potential management options regarding public health policies and strategies to prevent and manage future outbreaks.

An important strategy to manage the impact of acute infections and prevent the long-term effects of COVID-19 in the first place is vaccinations. Some previous research on SARS-CoV-2 has shown that vaccinations can help to reduce infection severity or prevent severe symptom developments over time [19]. This can also lower the risk of hospitalization and death. In a systematic review, however, insufficient evidence (e.g., from cohort and case-control studies) was found that vaccination before the acute SARS-Cov-2 infection can reduce the subsequent risk of developing post-/long-COVID [20]. 

Other research compared individuals with breakthrough SARS-CoV-2 infections (BTI; i.e., individuals who were previously vaccinated) with people who had SARS-CoV-2 infections and who were not vaccinated using an extensive US database [21]. The researchers found that BTI had lower mortality and morbidity risk (post-acute sequelae), which shows the usefulness of vaccinations against a SARS-CoV-2 infection. 

Accordingly, vaccination before infection appears to be a partial protection in the post-acute phase of the disease [21]. However, the current evidence is ambiguous, as not all previous studies found a relationship between post-/long-COVID and vaccination status [22], but rather individual behavior prior to the SARS-Cov-2 infection [23], former social-cognitive beliefs [13,24] or pre-existing cognitive and neurological vulnerabilities [25]. 

The effect of the vaccination concerning the timepoint relating to the acute SARS-Cov-2 infection is largely unclear. Additionally, the relationship between the vaccination after developing post-/long-COVID symptoms remains unclear [20]. Further research [26] showed no change in post-/long-COVID symptoms after an additional vaccination. This experience can persuade patients not to uptake additional vaccinations. However, there are also other potential reasons why people do not take part in vaccinations. For example, it was reported that the main reasons against a second vaccine were the fear of worsening symptoms (55.9%) and the belief that the vaccination was not indicated after the manifestation of post-/long-COVID symptoms [27]. 

Different theories, such as the compensatory carry-over action model and health beliefs, explain irrational decisions by considering different facets [13,24]. This is imperative for adequate communication strategies that provide objective information about the usefulness of vaccinations and when to get vaccinated. Taking vaccination status into account is crucial to target the psychological determinants of behavior [23]. Additionally, previous studies called for tailored immunization campaigns as rewards appear relatively ineffective in motivating unvaccinated individuals but could motivate those with previous vaccination attempts [23], especially if they are convinced about their effectiveness. 

Moreover, it was discovered that specific SARS-Cov-2 vaccinations could be associated with more physio-affective post-/long-COVID symptoms, that is, fatigue, anxiety, depression, and an autoimmune response [28]. Thus, it is essential not only to look at physical symptoms but also at affective and neurological impairments. Equivalent to the latter, other research found effects in post-/long-COVID patients, i.e., that social participation, workability, and life satisfaction were decreased even if patients were vaccinated [29]. The results seem to depend on the type of vaccination, number of doses, and timepoints concerning the acute SARS-CoV-2 infection. 

Regarding the type of vaccination, being vaccinated 2+ times with specific vaccinations (BNT162b2, BioNTech/Pfizer) was associated with a decreased risk of reporting post-/long-COVID symptoms [30]. The researchers who conducted this study concluded that the vaccination could have a protective impact on symptoms. However, it is unclear what caused this protective impact—vaccination with BNT162b2 or the study methodology, i.e., that a selective sample was drawn of only RT-PCR positively tested patients in three major government hospitals in Northern Israel between July and November 2021. Additionally, the authors compared the group who received more than two vaccinations to unvaccinated and not previously infected groups. 

The descriptive differences between vaccinated once and infected vs. those without vaccination appear pessimistic. Only if patients were vaccinated twice did a protective effect occur. Following this line of thought, the number of doses seems relevant for the protective effects of vaccinations against post-/long-COVID. In some studies, participants had more severe symptoms after a single vaccination than participants who received more doses, with no differences for the vaccination type [27,31,32]. 

Following the gaps in current literature, examined post-/long-COVID patients’ vaccination status and the timepoint of their infection regarding different outcomes, namely social participation, workability, and life satisfaction over time. In previous research, higher age, female sex, and the severity of the acute illness were associated with a worse trajectory of post-/long-COVID [33], so these variables should be statistically controlled. In more detail, we aim to answer the following *research questions*:(1)How many patients positively screened with post-/long-COVID were vaccinated once or completely (2 or 3 times)?(2)Is the timepoint of vaccination (before vs. during/after the COVID-19 infection) related to the post-/long-COVID symptom severity development over time?(3)To what extent do differences in post-/long-COVID patients’ functional status (i.e., perceived symptom severity, social participation, workability, and life satisfaction) relate to whether they are vaccinated?

## 2. Materials and Methods

### 2.1. Participants and Procedure

The current study with patients suffering from post-/long-COVID symptoms was a state-wide, regional online survey in Bavaria (Germany). Patients were recruited for an intervention study targeting the patients’ health and well-being. Recruitment was performed via medical doctors, outpatient clinics, social media, and press releases. 

After providing informed consent at the T1 measurement point (baseline), participants answered the survey questions. At the end of the T1 questionnaire, all participants were thanked and were given the opportunity to leave their e-mail addresses on a different website to remain in the study. The follow-up surveys took place 3–4 (T2) and 4–5 (T3) weeks after the initial screening survey at baseline (see Figure 1). 

The patients were provided a “personal pilot” for individual contacts, and they were enrolled in a structured study program [34]. Additionally, they received personal coaching sessions with their personal pilot to learn how to utilize the German healthcare system better and get support to deal with their symptoms in everyday life, return to work, and improve their health and well-being sustainably [34]. The personal pilots also interviewed the participants regarding their vaccination status. The pilots reminded the patients to participate in the online surveys from which we analyzed the data to test the research questions.

Participants remained in the study if they screened positive for post-/long-COVID and met the study criteria (see Figure 1 for the drop-out flow chart). To determine whether participants were eligible to take part in the study, patients were asked to fill out a screening tool with symptoms of post-/long-COVID described in a previous publication [34]. Within this screening, the criteria to be diagnosed with post-/long-COVID were: (1) having at least three out of 14 common post-/long-COVID symptoms (such as fatigue, brain fog, etc.) that are rated with a severity greater than 3 (on a scale of 1 = no problem to 5 = extreme problem); (2) which must not be explainable by the patient by other causes. Furthermore, (3) the symptoms should either be new or existing and should have worsened due to the COVID-19 infection. 

Additionally, (4) the COVID-19 infection must have been more than four weeks ago, and (5) out of four options that ask for the overall improvement of the general state of health, patients were excluded if they indicated that they are already doing well and that they are optimistic that they would fully recover soon. Moreover, (6) regarding their daily lives, patients were included only if they indicated limitations related to their post-/long-COVID symptoms. Furthermore, the screening assessed whether patients (7) had received previous treatment such as medical rehabilitation and (8) worked in health care/laboratory (because these individuals get special health care treatment in Germany). Patients were excluded from the study if criteria 7 or 8 were positive. 

The included participants were predominantly female (71.9%) and aged *M* = 44.19 years (*SD* = 11.84, range: 20–61). Data were collected between March and October 2022 using the survey tool UNIPARK. Personal pilots worked with a regular landline or mobile phone and video conferencing systems (MS TEAMS).

### 2.2. Measurements

*Symptoms* were measured at all timepoints [34]. Participants were asked to indicate their symptoms. They were presented with 14 common post-/long-COVID symptom categories (such as fatigue, brain fog, etc.) and were asked to rate them on a scale of 0 = no problem to 3 = extreme problem. An overall mean scale was calculated.

*Life satisfaction* was assessed with four questions. One example question is, “How satisfied are you with your life in general at the moment?” [35,36]. Participants could answer on a scale from 1 (very unsatisfied) to 5 (very satisfied). Cronbach’s alpha was 0.731.

*Social participation* was measured using a 12-item questionnaire [37]. The questionnaire asked how participants were able to perform specific activities over the past week (e.g., doing things that require the use of hands and fingers, such as picking up small objects or opening containers, etc.). Participants could answer these questions on a scale from 1 (never) to 5 (always). Cronbach’s alpha was 0.891.

*Workability* was measured with one item from the WAI “Assume that your workability at its best has a value of 10 points: How many points would you give for your current workability?” Patients could answer these questions on a scale from 0 (means that currently cannot work at all) to 10 (work ability at its best) [38].

The *vaccination status* was assessed via computer-assisted telephone interviews (CATI). The interviewer (personal pilot) asked the patient openly for his/her vaccination status and coded the answers into the following scheme: 1 = No, 2 = Yes, but with a vaccine that has not been licensed in the EU, 3 = Yes, with the Johnson and Johnson vaccine, 4 = Yes, with the Johnson and Johnson vaccine plus another vaccination, 5 = Yes, with another vaccine that has been licensed in the EU, e.g., from BioNTech, but only once, 6 = Yes, twice with another vaccine that has been approved in the EU, e.g., from BioNTech, 7 = Yes, I have had more than two vaccinations, 8 = Other. 

The *timepoint of infection* was assessed during the CATI. The interviewers coded the answer to the question when the patient was infected—before vaccination, at the same time as the vaccination or more than two weeks after vaccination.

*Socio-demographics* such as sex and age were measured by self-report in the online questionnaires.

### 2.3. Data Analysis

To test *RQ1* (How many patients positively screened with post-/long-COVID were vaccinated once or completely (2 or 3 times)?), we conducted frequency analyses. To test *RQ2* (Is the timepoint of vaccination [before vs. during/after the COVID-19 infection] related to the post-/long-COVID symptom severity development over time?), repeated measurement analysis of variances (RM ANOVA) was conducted using the timepoint of the infection (before vs. during or after the vaccination) as the independent variable and symptom severity as a repeated measurement factor. Lastly, to test *RQ3* (To what extent do differences in post-/long-COVID patients’ functional status [i.e., perceived symptom severity, social participation, workability, and life satisfaction] relate to whether they are vaccinated?), we conducted a MANOVA to test whether the reported time of infection (T1) was related to the functional status of individuals (i.e., their symptom severity, social participation, life satisfaction, and workability) at T2. The analyses took the sex and age of individuals as covariates.

## 3. Results

### 3.1. Testing RQ1: How Many Patients Positively Screened with Post-/Long-COVID Were Vaccinated Once or Completely (2 or 3 Times)?

Regarding RQ1, results show that of those who provided information about their vaccination status (*n* = 225), most participants were vaccinated at least once (*n* = 170, 75.55%), and only *n* = 8 (3.56%) were not vaccinated at all, see Table 1.

Recapping findings regarding RQ1, three out of four patients positively screened with post-/long-COVID were vaccinated once or completely (2 or 3 times), and one out of five did not indicate their vaccination status. The minority indicated not being vaccinated; every fifth patient (20%) indicated being vaccinated twice, and every second participant indicated being vaccinated three times (53.3%). The descriptives for patients with different vaccination statuses are presented in detail in the Appendix A.

### 3.2. Testing RQ2: Is the Timepoint of Vaccination (before vs. during/after the COVID-19 Infection) Related to the Post-/Long-COVID Symptom Severity Development over Time?

Regarding the timepoint of the vaccination, of those who answered this question (*n* ≥ 172), *n* = 43 (25%) of the patients were vaccinated before their infection, *n* = 11 (6.4%) were vaccinated around the same time as their acute infection, and the majority of the study participants, i.e., *n* = 118 (68.6%) received their vaccinations more than two weeks before they got infected.

An RM ANOVA was conducted by considering symptom severity over time as the dependent variable and the timepoint of the infection (before vs. during/after the vaccination) as a between-subject factor. Results reveal a main effect of time *F* (2,270) = 50.96, *p* < 0.001, ƞ_part_^2^ = 0.274. Repeated contrast effects show that for the study sample answering the question regarding vaccination status, the symptoms decreased significantly from T1 to T2 with *F* (1,135) = 59.45, *p* < 0.001, ƞ_part_^2^ = 0.306, but not between T2 and T3 with *F* (1,135) = 1.93, *p* = 0.167, ƞ_part_^2^ = 0.014. Furthermore, the RM ANOVA shows no main effect of the timepoint of the infection with *F* (1,135) = 0.55, *p* = 0.459, ƞ_part_^2^ = 0.004, but an interaction of time and timepoint of the infection *F* (2,270) = 3.61, *p* = 0.028, ƞ_part_^2^ = 0.026. The results are reported in Figure 2. Means and standard deviations are displayed in the Appendix A.

Summarizing findings regarding RQ2, the timepoint of vaccination (before vs. during/after the COVID-19 infection) interacted with the main trend over time. Those patients getting infected before getting vaccinated reported higher symptom severity than the ones being infected during or after their vaccination. This difference was only significant at the measurement point T1 but not at T2 or T3. However, the symptoms decreased significantly over time in both groups. 

### 3.3. Testing RQ3: To What Extent Do Differences in Post-/Long-COVID Patients’ Functional Status (i.e., Perceived Symptom Severity, Social Participation, Workability, and Life Satisfaction) Relate to Whether They Are Vaccinated?

To test RQ3, a MANOVA was conducted with the vaccination status and sex of the participant as the independent variable, mean symptom severity at T2, social participation at T2, life satisfaction at T2, and workability at T2 as the dependent variables, and age as a covariate (The MANOVA was replicated by additionally including BMI as a covariate validating the results and revealing no significant effect of BMI.). The results are displayed in Figure 3. The multivariate tests show that the overall effect of vaccination status was significant with *F* (16,596) = 2.29, *p* = 0.003, ƞ_part_^2^ = 0.058. No other multivariate test revealed significance with *p ≥* 0.121. 

Single comparisons revealed that there were significant differences in *life satisfaction* between the different groups with *F* (4,149) = 3.89, *p* = 0.005, ƞ_part_^2^ = 0.094 and *workability* with F [4,149] = 4.10, *p* = 0.003, ƞ_part_^2^ = 0.099. However, there were no further differences, neither regarding social participation (F [4,149] = 1.78, *p* = 0.136, ƞ_part_^2^ = 0.046) nor *symptom severity* (*F* [4,149] = 0.07, *p* = 0.990, ƞ_part_^2^ = 0.002).

To understand the relationship between the frequency of vaccination and life satisfaction and workability better, we first ran bivariate correlation analyses with vaccination status from 0 (no vaccination) to 3 (three or more vaccinations) and life satisfaction, as well as workability. The results revealed that vaccination status and life satisfaction, as well as vaccination status and workability, are negatively correlated with *r* (156) = −0.311, *p* < 0.001, and *r* (155) = −0.278, *p* < 0.001, respectively. This demonstrates that the patients being vaccinated more often reported more limited life satisfaction and workability. 

To check the relationships further, we examined which groups’ life satisfaction statistically differs. Thus, we ran a univariate ANOVA with vaccination status as the independent variable and life satisfaction as the dependent variable. The main effect of vaccination status was again significant with *F* (1,156) = 4.87, *p* < 0.001, ƞ_part_^2^ = 0.111. The Bonferroni post hoc test shows significant differences between individuals vaccinated twice (*M* = 2.45, *SD* = 0.50) compared to those vaccinated three or more times (*M* = 2.07, *SD* = 0.63, *p* = 0.021). Additionally, the patients vaccinated twice differed significantly from those who did not indicate their vaccination status (*M* = 1.75, *SD* = 0.62, *p* = 0.043).

We used a similar ANOVA as described above to check between-group differences in workability. The results showed a main effect of vaccination status with *F* (1,155) = 3.88, *p* = 0.005, ƞ_part_^2^ = 0.091 again. Here, Bonferroni post hoc analyses show no differences between the groups with *p* ≥ 0.057. Descriptive statistics for the MANOVA are reported in the Appendix A.

Exploratory qualitative analyses from the telephone interviews revealed that four of the eight patients not being vaccinated had consulted an alternative practitioner for their symptom treatment. Those patients reported receiving alternative medical treatment against their symptoms, such as taking globules, natrium chloratum, infusions, “ampoules”, and vitamin B.

Concluding findings regarding RQ3, there were significant differences in post-/long-COVID patients’ life satisfaction and workability regarding whether they were vaccinated. No differences concerning vaccination status were revealed regarding perceived symptom severity and social participation.

## 4. Discussion

This study incorporated 225 post-/long-COVID patients and sought to understand the relationship between their vaccination status and functional status (workability, social participation, and life satisfaction) over time. In doing so, we examined how many post-/long-COVID patients were vaccinated and whether the timepoint of infection (i.e., before vs. during/after the vaccination) was related to the symptom severity over time. It also investigated whether there were differences in the functional status of patients depending on whether they were vaccinated once, twice, or three times, or not vaccinated at all. 

The results are discussed based on the current literature to give implications for future research and to prevent and overcome pandemics and the aftermath of the COVID-19 pandemic, i.e., post-/long-COVID. However, it should be considered that some patients did not answer the question regarding their vaccination status, which could have been caused by different reasons (e.g., some were not vaccinated but were not willing to report this). This aligns with previous research results, indicating that social-psychological mechanisms such as intergroup mechanisms can occur. Vaccination status seems to change the perspective of intergroup situations, and participants perhaps wanted to avoid stigmatization or discrimination if they revealed that they belonged to the “outgroup” of non-vaccinated people [39]. 

However, this group of non-responders was intentionally added to the analytical procedure to inspect their symptom severity, social participation, life satisfaction, and workability compared to all other groups. In the past, it was found that individuals with specific beliefs may subjectively rate the benefits of vaccination as insufficient in comparison to other behaviors [24]. To keep all patients in the picture, even if they do not provide data regarding their vaccination status, it is vital not to exclude them. However, it would be interesting to investigate this group further, especially as their life satisfaction appeared lowest and their workability was reported as limited compared to those who stated being not vaccinated.

Regarding the findings of this study, we found that almost all post-/long-COVID patients who indicated their vaccination status were vaccinated at least twice, and a majority received more than two vaccinations. Only 3.5% of patients who indicated their vaccination status reported not being vaccinated. As of 9 January 2023, the basic immunization in Germany was 75.1% in Bavaria [40]. Even though the official authorities (the Robert Koch Institute, RKI) assume that actual rates might be up to 5% higher, it is still below the recommended vaccination rate and the level in the sample of this study. If we assume that all patients who did not indicate their vaccination status also were not vaccinated, the rate in this sample of 77.6% is comparable to the overall vaccination rate in Bavaria. Nevertheless, all participants in this study suffered from post-/long-COVID (as this was an inclusion criterion to become a study participant). The current study results indicate that even with high immunization rates, the long-term symptoms of the infection still pose a critical challenge to the healthcare and insurance systems. 

About two-thirds (68.6%) of the participants in this study received three vaccinations before they were infected. This aligns with previous studies that found no clear relationship between post-/long-COVID and the previous vaccination status [22]. On the other hand, previous evidence (e.g., [20]) concluded from six articles that vaccination could reduce the risk of post-/long-COVID, suggesting that every additional dose is more effective (see also [41,42]) in reducing the risk for post-/long-COVID. Overall, there is an agreement [2,3,4,5,6,7,8,9,15,16,17,18,19,20,21,22] that vaccination cannot entirely avoid the risk of breakthrough infections and, thus, long-term symptoms. 

Our study only looked at patients suffering from post-/long-COVID and found differences regarding the timepoint of the vaccination only at the baseline measurement point. However, symptom severity decreased in both groups at the follow-up measurement points, and no differences were detectable at the follow-up measurement points. This general trend of symptom trajectory in our study complies with previous findings regarding a likely recovery overall [3,14,31,43,44,45,46,47,48]. On the one side, the trend in symptom reduction is positive, i.e., symptom reduction takes place over time. On the other side, the question remains why patients with at least two vaccinations appear more limited in their life satisfaction. In our sample, most individuals were vaccinated at least twice, and alarmingly, their life satisfaction was lower than all other vaccinated groups. 

Surprisingly, workability was also negatively correlated with vaccination status. It is possible that the participants in our sample perceived a higher need for another vaccination after experiencing higher symptoms, which aligns with the finding that people who were infected before their first vaccination show higher symptoms. Other studies also discussed whether lower symptom severity scores are related to the vaccination status, indicating that specific SARS-Cov-2 vaccinations can worsen physio-affective post-/long-COVID symptoms [28]. However, the patients in this study did not differ significantly regarding their symptom severity and social participation based on their vaccination status. Accordingly, Wisnivesky et al. found no relationship between vaccination status and mental health or quality of life [22]. This contrasts previous research in which vaccination was associated with fewer reported symptoms [22]. Nevertheless, the association between vaccination status and long-/post-COVID symptoms or mental health status has not been sufficiently investigated and needs more attention in the future. 

As indicated above, patients who were infected before receiving their first vaccination also reported higher symptoms at T1 than the ones who were infected at the same time or after their vaccination. This underlines the potential of vaccination as a preventive measure, especially since there was a significant interaction effect between the timepoint of vaccination and symptom reduction over time. This result is in line with a previous literature review which has found very heterogenous results from improvements over stagnation to worsening of post-/long-COVID conditions [20].

Future interdisciplinary research needs to establish whether there are biological, behavioral, and psychological predictors that determine the effect of vaccination after infection on functional immune response and recovery processes [23,25]. Although both cognitive behavioral therapy and graded exercise therapy appear rather ineffective and have potential side effects, some lifestyle changes, such as plant-based diets, have been found to be promising in supporting post-/long-COVID management [49,50]. 

Consequently, vaccinations could be more targeted to alleviate symptoms and support post-/long-COVID patients, combined with these other disease prevention measures and management strategies [5]. It seems especially important to give such patients confidence that there are active options for managing and improving their post-/long-COVID symptoms severity (e.g., [2,3,6,8]), to return to work and social life and to regain life satisfaction.

### Limitations and Future Research Directions

Open questions remain from this study, such as whether the type of vaccination would impact the outcomes. While we collected this basic information, we focused on the number of doses to ensure statistical power. Regarding the analyzed number of doses and timepoints, the results are ambiguous. In future studies, systematic research is needed regarding:the number and type of vaccinations,comparing samples of people without post-/long-COVID symptoms but various vaccination states,replicating the results with those from larger patients’ samples [27,28,29,30,31,32], orvalidate the current findings with studies with samples from other states and countries, cultures, and continents, as the effects we found might have been specified for this cohort.

Previous studies tested a selective sample with only RT-PCR positively tested patients in three major government hospitals in Northern Israel in 2021 [30]. In comparison, our study contained positively screened patients based on self-reported data in Bavaria, Germany, who were recruited in 2022. Whether our results differ from those of other researchers due to these aspects remains open and calls for systematic reviews and meta-analyses. 

This study is limited by its uncontrolled nature. In future studies, the patients with long-/post-COVID should be compared to matched patients without such a disease trajectory but who rather recovered and tested for the effect of the different vaccinations. Ideally, prospective data should be used for that. Furthermore, we need more experimental studies in which different interventions help people decide on indicative vaccinations [24] and recover from symptoms and constraints [34]. Potential vicious circles and maladaptive processes should be addressed [12], and corresponding interventions need testing in randomized controlled trials.

Moreover, studies should validate the subjective nature of the measures with objective assessments to overcome biases and measurement errors. Finally, more in-depth interviews can improve understanding of patients’ demands and constraints and develop interventions that are well-matched to patients’ needs. This would be especially needed to understand and address the reasons individuals decide against vaccination, including the fear of worsening symptoms and the belief that the vaccination was not indicated after the manifestation of post-/long-COVID symptoms [27]. 

Research on the connection between immunological findings and the severity of long-/post-COVID [50,51] calls for future investigations since studies suggest that a novel phenotype of proliferative-exhausted CD8-T-Cells is strongly associated with the severity of acute COVID-19 [52]. Furthermore, long-/post-COVID could be exacerbated by autoimmune-related self-reactive T cells, which may be able to control KIR+ CD8 T cells [51]. Moreover, previous studies (such as [52,53]) investigated the connections between levels of SARS-CoV-2 antibodies and autoantibodies, as well as post-/long-COVID. Findings (e.g., [26,51]) demonstrate that the antibody levels of the patients change after vaccination. In the current study, the antibody titer information was not assessed and, accordingly, cannot be analyzed. Future research should investigate whether the antibody level induced by vaccination differs from the one due to the real virus infection to validate earlier research (e.g., [52,53]) and fill this gap with the current study, as antibody titer data were unavailable. 

Implications to prevent post-/long-COVID can only partially be drawn from this study. However, reviewing the current evidence, we can conclude that vaccinations are just one strategy to prevent severe infections and the long-term, post-acute sequelae of infection [26]. Attempts to overcome post-/long-COVID should take more options than vaccinations into account, such as behavioral interventions and medical rehabilitation treatments addressing the coping mechanisms, and appropriate, safe patient behavior is required [13,24,34]. This is especially important because limited workability is problematic not only for the individual (because of limited social participation and economic demands) but also for society (because of limited productivity and skills shortage, and costly reintegration demands to be covered by the social welfare system). This needs to be addressed based on theories and evidence, including psychological and behavioral approaches (e.g., [49]), immune phenotypes [54], and adaptive-like NK cell phenotype [55,56]. 

## 5. Conclusions

Most of the patients in this study were vaccinated at least twice. However, approximately one out of five did not disclose their vaccination status. While the vaccination status and timepoint concerning the infection were not related to symptom severity over time, many vaccinated patients still suffer from post-/long-COVID in terms of limited life satisfaction and workability. The latter is important for the individual but also the society. Vaccination cannot entirely prevent the aftermath of the pandemic in terms of breakthrough infections and post-/long-COVID; it can just be a part of different primary and secondary prevention measures working as a general strategy in terms of public health.

Vaccinations should be available and accessible to the whole population to improve herd immunity. Additionally, there is a need for a communication strategy providing information about the usefulness of vaccination, especially as our study has shown that symptom severity was lower in those patients who were vaccinated prior to infection compared to those getting infected prior to or at the same time of the vaccination. However, the finding that being vaccinated was more often correlated with lower life satisfaction, and lower workability requires more attention.

## Figures and Tables

**Figure 1 vaccines-11-00691-f001:**
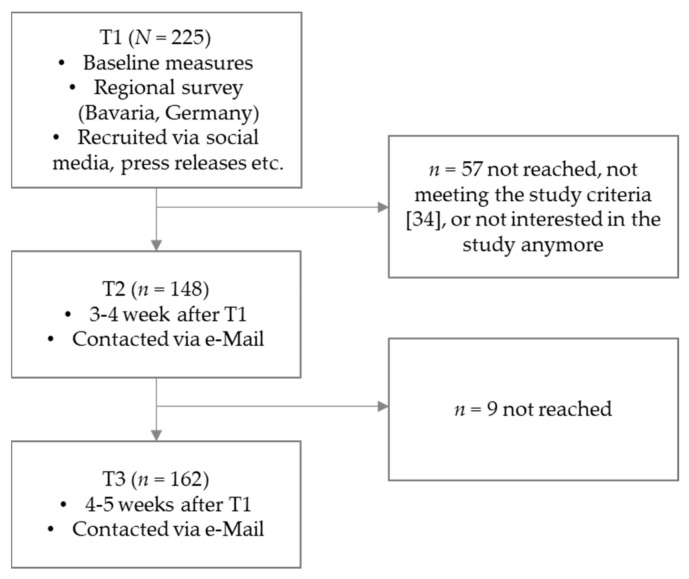
Flow chart with study participant flow over the three measurement timepoints [34].

**Figure 2 vaccines-11-00691-f002:**
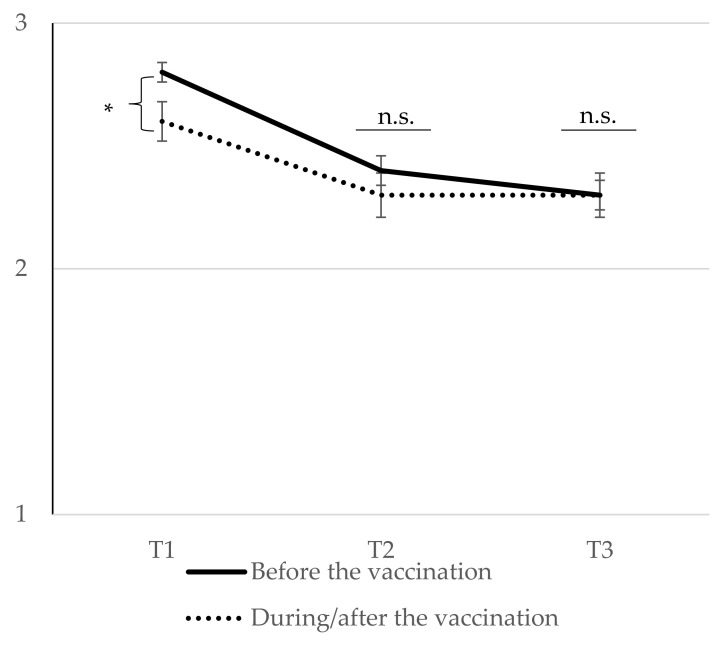
Symptom severity at T1, T2, and T3 as a function of the timepoint of the infection. Note. *N*_T1_ = 225, *N*_T2_ = 167, *N*_T3_ = 159, *N*_T1-T3_ = 137; * *p* = 0.017, n.s. = not significant, error bars represent standard errors of the means (To examine differences in symptom severity at certain timepoints, we ran three independent *t*-tests with the timepoint of infection as the independent variable and symptom severity at T1, T2, and T3, respectively, as the dependent variable. The results show that perceived symptom severity differs at T1 between the groups with *t* (169) = 2.40, *p* = 0.017, *d* = 0.426, but not at T2 with *t*(149) = 0.35, *p* = 0.972, *d* = 0.006, or at T3 with *t* (140) = 0.138, *p* = 0.890. *d* = 0.027.).

**Figure 3 vaccines-11-00691-f003:**
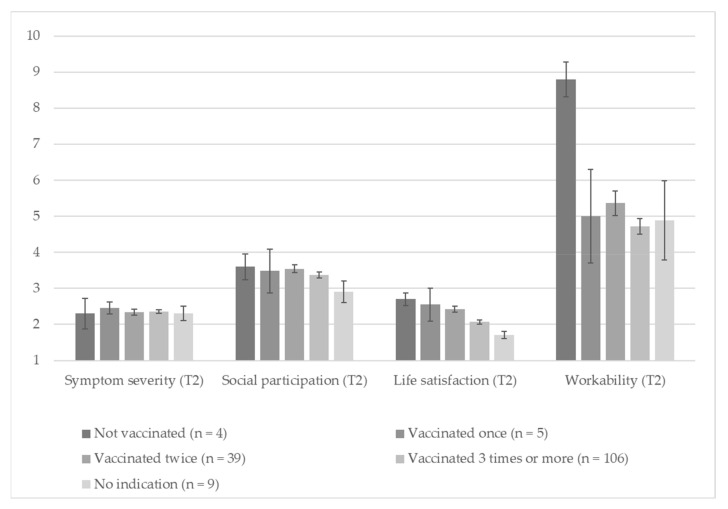
The relationship between patients’ self-reported vaccination status and their perceived functional status indicated by symptom severity, social participation, life satisfaction, and workability. Note. Error bars represent standard errors of the mean scores.

**Table 1 vaccines-11-00691-t001:** Vaccination status.

Vaccination Status	*n*	%	Cumulative %
0 Not vaccinated	8	3.56	3.56
1 Vaccinated once	5	2.22	5.78
2 Vaccinated twice	45	20.00	25.78
3 Vaccinated ≥3 times	120	53.33	79.11
4 Not indicated	47	20.89	100.0
Total	225	100.0	

Note. Group 4 did not provide any information on their vaccination status.

## Data Availability

The data presented in this study are available on request from the corresponding author. The data are not publicly available due to confidential patient information being used.

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
