# Peer review of "Patients’ Post-/Long-COVID Symptoms, Vaccination and Functional Status—Findings from a State-Wide Online Screening Study"

_vaccines, 2023, doi:10.3390/vaccines11030691_

Round 1

Reviewer 1 Report

This research is about the relationship between long-COVID symptoms and vaccination status. Two main results are concluded: (1) the vaccination timepoint was not related to the long-COVID symptoms severity over four weeks; (2) the vaccination status was not related to the perceived functional status (symptom severity, social participation, life satisfaction and workability) of long-COVID patients. Although the research about whether vaccination can release long-COVID symptoms has been reported by dozens of studies, this manuscript provided more information. However, the study need to resolve some questions.

1) The expression in the manuscript need to be carefully checked to avoid confusion. For example, what is the exact number of patients that had been vaccinated before SARS-CoV-2 infection (line 224-225)?

2) In the discussion, authors wrote that patients who were infected before receiving their first vaccination reported higher symptoms at T1 than the ones who were infected at the same time or after their vaccination. The result was derived from Figure 2. In Figure 2, the significance analysis of the difference at T1 and T2 in 2 groups are required.

3) Why authors only present relationship between patients’ vaccination status and their perceived functional status at T2? What about their relationship in T1 and T3?

Author Response

Comments and Suggestions for Authors

This research is about the relationship between long-COVID symptoms and vaccination status. Two main results are concluded: (1) the vaccination timepoint was not related to the long-COVID symptoms severity over four weeks; (2) the vaccination status was not related to the perceived functional status (symptom severity, social participation, life satisfaction and workability) of long-COVID patients. Although the research about whether vaccination can release long-COVID symptoms has been reported by dozens of studies, this manuscript provided more information. However, the study need to resolve some questions.

AUTHORS’ RESPONSE: Thank you very much, we appreciate this great summary.

1) The expression in the manuscript need to be carefully checked to avoid confusion. For example, what is the exact number of patients that had been vaccinated before SARS-CoV-2 infection (line 224-225)?

AUTHORS’ RESPONSE: We are very sorry for the typos, we carefully checked all expressions and numbers again and corrected typos (see the version of the manuscript with changes tracked).

2) In the discussion, authors wrote that patients who were infected before receiving their first vaccination reported higher symptoms at T1 than the ones who were infected at the same time or after their vaccination. The result was derived from Figure 2. In Figure 2, the significance analysis of the difference at T1 and T2 in 2 groups are required.

AUTHORS’ RESPONSE: Thanks very much, we added this information to Figure 2.

3) Why authors only present relationship between patients’ vaccination status and their perceived functional status at T2? What about their relationship in T1 and T3?

AUTHORS’ RESPONSE: Thank you for your comments on our manuscript. We are sorry for the confusion and can assure that we assessed symptom severity over time (see Figure 2 with symptom severity at T1, T2 and T3). Accordingly, we amended the figure caption to “Figure 3. The relationship between patients’ self-reported vaccination status and their perceived functional status indicated by symptom severity, social participation, life satisfaction and workability.” (lines 295ff).

Workability, social participation, and life-satisfaction were only assessed at T2 and not at T1 or T3. The reason for this was that T1 was designed as a short screening assessment, which was intended to be as brief as possible (as noted in Dahmen et al., 2022) because the patients piloted appeared very limited with the capacity to answer questions and we did not want to burden them further. Additionally, we did not want to use the same questions again for participants who were surveyed again at T3, which came shortly after T2.

We understand your concern about the inability to address the relationship between patients’ vaccination status and their perceived functional status at all three time points. However, given the reasons outlined above, we were not able to collect data on these measures at T1 and T3. We believe that the data we have collected at T2 appears sufficiently meaningful and gives valuable insights into the research questions that we outlined within the manuscript.

Reviewer 2 Report

1. Title: good
2. Abstract: it captures the appropriate essence of the manuscript. Excellent.
3. Introduction: The introduction identifies the problem that is being addressed in the manuscript and develops and states the purpose of the manuscript.

4. References: I have verified all references and all key references are correct

5. Methods: 223 patients is not appropriate for this kind of research. Why this limitation?

6. Discussion:

* The authors  discussed the  limitations of the study. I appreciate it
7. Conclusion: The conclusion is justified by the methods and results.

Author Response

Comments and Suggestions for Authors

  1. Title: good
    2. Abstract: it captures the appropriate essence of the manuscript. Excellent.
    3. Introduction: The introduction identifies the problem that is being addressed in the manuscript and develops and states the purpose of the manuscript.
    4. References: I have verified all references and all key references are correct
  2. Methods: 223 patients is not appropriate for this kind of research. Why this limitation?
  3. Discussion: * The authors  discussed the  limitations of the study. I appreciate it
    7. Conclusion: The conclusion is justified by the methods and results.

AUTHORS’ RESPONSE: Thank you very much for your feedback #1-4 and #6-7. Regarding #5, we appreciate your question and would like to elaborate on this point by highlighting that we randomly recruited patients throughout Germany and accordingly titled our manuscript “…State-wide Online-Screening Study”. While we agree that a larger sample would be stronger, we tried to recruit a representative sample of long-COVID patients and state accordingly “The current study with patients suffering from post-/long-COVID symptoms was a state-wide, regional online survey in Bavaria (Germany). Patients were recruited for an intervention study targeting the patients' health and wellbeing. Recruitment was performed via medical doctors and outpatient clinics as well as social media and press releases.” (lines 142ff).

To take the sample size into account, we report effect sizes (ƞpart2 and d).

Moreover, we added this point to the discussion “In future studies, systematically the number and type of vaccinations should be tracked and compared with larger patients’ samples [27–32] or also samples of people without post-/long-COVID symptoms but various vaccination states. Also, in general, studies with representative samples should validate the current findings, as the effects we found that this might have been specific for this cohort. (lines 444ff).

Furthermore, we corrected the report to “This study examined 225 post-/long-COVID patients…“ (line 339) as we were able to include 2 more study participants.

Reviewer 3 Report

Lippke et al investigate the association of vaccination and long covid via a contry-wide online-screening study. They find that timepoint of vaccination was related to symptom severity at t1 and being vaccinated was associated with lower life satisfaction at T2. Overall, the problem that the authors try to address is timely and important. However, the findings from this current analysis is very shallow and is not interesting enough. Meanwhile, most of the discussion from this paper is limited and many important aspects are missing. Thus, the reviewer recommend at least a major revision before it can be further considered.

1.       It will be important to validate some of the findings by analyzing some published paper’s data to confirm the findings from this paper are not cohort specific.

2.       Lack of information on how to account for other confounding factors when performing the analysis. It was not clear how did the authors statistically account for other confounding factors (e.g. gender, age, BMI), in many of the key analyses? If haven’t, the authors should account for these confounding factors and re-evaluate if any of these findings are still significant.

3.       Most of the discussion contents are shallow. More in-depth discussion connecting some of more novel findings from literature as future directions can further elevate the depth of the current manuscript. For example, it will be important to investigate what are the specific immune cell phenotypes that contributed or “anti-contribute” to long covid pre and post vaccination. There are many recent literatures that give specific examples of a few novel immune phenotypes that has been identified in COVID-19. Those specific novel immune phenotypes could be investigated or at least specifically discussed within the context of this study. For example, there has been recent reports of proliferative-exhausted CD8 T cells (PMID: 33171100) which seems to be a novel phenotype that are strongly associated with COVID-19 severity. Will it also be potentially associated with long covid? Discussion of their role within the context of long covid connection as a future direction of investigation could improve the depth of the current manuscript. Similarly, there has been reports (PMID: 35258337) on KIR+ CD8 T cells that controls self-reactive autoimmune T cells and has been reported in COVID. A natural future direction is to connecting this novel phenotype to long covid, since long covid could be exacerbated by autoimmune related self-reactive T cells, which may be able to control by KIR+ CD8 T cells.

4.       Engaging the contents of these novel immune phenotypes of COVID-19 with long covid as future research directions within the context of recent literatures will further strength the depth of the manuscript and make it more interesting for the audience to read. One example is that there has been recent literatures (e.g. PMID: 34489601) reporting that monocytes bifurcated into two subpopulations pro-inflammatory and anti-inflammatory subpopulations and they found the quality and quantity of proinflammatory subsets is increased with covid-19 severity. Will any of these monocytes be contributing to the long covid differences? This can be discussed to further elevate the depth of this paper. Similarly, there has already been recent literatures on the novel “adaptive NK phenotype” (PMID: 36066491) and on NK cell with unique early IFNa signatures (PMID: 34592166). These two NK phenotypes are both closely associated with covid-19 severity. It will be one of the important future directions to further evaluate these NK phenotypes in the future long cohorts to study these novel phenotypes’ association with protections.

5.       Accessibility of de-identified data. The data collected from this paper can be informative to the field of long covid research. It will be important for the authors to provide their de-identified data (e.g. symptoms status, vaccination timing, age, gender, etc. ) as supplementary tables. These data is helpful for the reviewers to evaluate their work, and this is also important later for the public to utilize these data to investigate their own hypothesis of interest since some of their more complex hypotheses could be beyond the current manuscript’ scope.

Author Response

Lippke et al investigate the association of vaccination and long covid via a contry-wide online-screening study. They find that timepoint of vaccination was related to symptom severity at t1 and being vaccinated was associated with lower life satisfaction at T2. Overall, the problem that the authors try to address is timely and important. However, the findings from this current analysis is very shallow and is not interesting enough. Meanwhile, most of the discussion from this paper is limited and many important aspects are missing. Thus, the reviewer recommend at least a major revision before it can be further considered.

AUTHORS’ RESPONSE: Thank you very much for giving us the chance to improve our paper. We included more previous evidence into the discussion, and improved the discussion to make it more profound and interesting. Please note that we mainly extended on psychological and behavioral aspects because this is the expertise we bring. For instance, we now report in the discussion “Different theories such as the compensatory carry-over action model and health beliefs explain such irrational decisions taking different facets into account [13,24]. This is imperative for adequate communication strategies providing objective information about the usefulness of vaccinations and when to get vaccinated. Taking the vaccination status into account is crucial to target the psychological determinants of behavior [49]. Also, previous studies were calling for tailored immunization campaigns as rewards appear rather ineffective to motivate unvaccinated individuals but could motivate those with previous vaccination attempts [49], especially if they are convinced about their effectiveness.” (lines 94ff).

  1. It will be important to validate some of the findings by analyzing some published paper’s data to confirm the findings from this paper are not cohort specific.

AUTHORS’ RESPONSE: Thank you, we agree that this is an important point and we have made sure that we do so, for instance with the following section on workability, life satisfaction and symptoms:

“…previous evidence (e.g., [20]) concluded from six articles that vaccination could reduce the risk of post-/long-COVID, suggesting that every additional dose is more effective (see also [41,42]) in reducing the risk for post-/long-COVID. Overall, there is an agreement [2-9, 15-22] that vaccination cannot entirely avoid the risk of breakthrough infections and, thus, long-term symptoms.

Our study only looked at patients who were suffering from post-/long-COVID and did find differences regarding the timepoint of the vaccination but only at the baseline measurement point. However, at the follow-up measurement points, symptom severity decreased in both groups and no differences were detectable at the follow-up measurement points any more. This general trend of symptom trajectory in our study complies with previous findings in terms of a likely recovery overall [3,14,31,43–48]. While the trend in symptom reduction is positive (i.e., that a symptom reduction takes place over time), at the same time, the question remains why patients with at least two vaccinations appear more limited in their life satisfaction: In our sample, most individuals were vaccinated at least twice, and alarmingly, their life satisfaction was lower than all other vaccinated groups.

Surprisingly, workability was also negatively correlated with vaccination status. It is possible that the participants in our sample perceived a higher need for another vaccination after experiencing higher symptoms, which is in line with the finding that people who were infected before their first vaccination showed higher symptoms. Whether lower scores are related to the vaccination status was also discussed in other studies indicating that specific SARS-Cov-2 vaccinations can worsen physio-affective post-/long-COVID symptoms [28]. However, the patients in this study did not differ significantly regarding their symptom severity and social participation based on their vaccination status. Accordingly, Wisnivesky et al. found no relationship between the vaccination status and mental health or quality of life [22]. This stands in contrast to previous research in which a vaccination was associated with less reported symptoms [30]. Nevertheless, the association between status of vaccination and long-/post-COVID symptoms, as well as mental health status is not yet sufficiently investigated. (line 386ff).

Additionally, we have list the following in the limitation section:

“Also, in general, studies with representative samples should validate the current findings, as the effects we found that this might have been specific for this cohort.“ (line 447ff) as well as

“Only 3.5% of patients who indicated their vaccination status reported not being vaccinated at all. As of January 9th, 2023, the basic immunization in Germany was 75.1% in Bavaria [40]. Even though the official authorities (the Robert Koch Institute, RKI) assume that actual rates might be up to 5 percentage points higher, it is still below the recommended vaccination rate as well as the level in the sample of this study. If we assume that all patients who did not indicate their vaccination status were not vaccinated, the rate in this sample (77.6%) is comparable to the overall vaccination rate in Bavaria.“ (lines 372ff)

And

“Moreover, this would also overcome the question of generalizability: Previous studies tested a selective sample with only RT-PCR positively tested patients in three major government hospitals in Northern Israel in the year 2021 [30]. In comparison, our study contained positively screened patients based on self-reported data in Bavaria, Germany, who were recruited in 2022. Whether our results are different from those of other researchers due to these aspects remains open at this point and calls for systematic reviews and meta-analyses.” (Lines 450ff).

  1. Lack of information on how to account for other confounding factors when performing the analysis. It was not clear how did the authors statistically account for other confounding factors (e.g. gender, age, BMI), in many of the key analyses? If haven’t, the authors should account for these confounding factors and re-evaluate if any of these findings are still significant.

AUTHORS’ RESPONSE: Thank you for this question. Indeed, we have accounted for gender and age, and now also for BMI, please see Footnote 2 on page 9 as well S1 and S3 in the appendix.

  1. Most of the discussion contents are shallow. More in-depth discussion connecting some of more novel findings from literature as future directions can further elevate the depth of the current manuscript. For example, it will be important to investigate what are the specific immune cell phenotypes that contributed or “anti-contribute” to long covid pre and post vaccination. There are many recent literatures that give specific examples of a few novel immune phenotypes that has been identified in COVID-19. Those specific novel immune phenotypes could be investigated or at least specifically discussed within the context of this study. For example, there has been recent reports of proliferative-exhausted CD8 T cells (PMID: 33171100) which seems to be a novel phenotype that are strongly associated with COVID-19 severity. Will it also be potentially associated with long covid? Discussion of their role within the context of long covid connection as a future direction of investigation could improve the depth of the current manuscript. Similarly, there has been reports (PMID: 35258337) on KIR+ CD8 T cells that controls self-reactive autoimmune T cells and has been reported in COVID. A natural future direction is to connecting this novel phenotype to long covid, since long covid could be exacerbated by autoimmune related self-reactive T cells, which may be able to control by KIR+ CD8 T cells.

AUTHORS’ RESPONSE: Thank you for your suggestions. We have expanded the discussion in more depth and connected some of more novel findings from literature as future directions. For instance, we now write “Research on the connection between immunological findings and the severity of long-/post-COVID [50, 52] calls for according for investigations in the future since studies suggest that a novel phenotype of proliferative-exhausted CD8-T-Cells is strongly associated with the severity of the acute COVID-19 [52]. Furthermore long-/post-COVID could be exacerbated by autoimmune related self-reactive T cells, which may be able to control by KIR+ CD8 T cells [53].

Implications to prevent post-/long-COVID can only partially be drawn from this study. However, reviewing also the current evidence provided by the literature, we can conclude that vaccinations are just one strategy to prevent severe infections and the long-term, post-acute sequelae of infection [26]. Implications to overcome post-/long-COVID should take more options than vaccinations into account: Behavioral interventions and medical rehabilitation treatments addressing the coping mechanisms and appropriate, safe patient behavior are required [13,24,34]. This is especially important because limited workability is problematic not only for the individual (because of limited social participation and economic demands) but also for society (because of limited productivity and skills shortage as well as costly reintegration demands to be covered by the social welfare system). This needs to be addressed on basis of theories and evidence, including psychological and behavioral approaches (e.g., [49]), immune phenotypes [50] and adaptive-like NK cell phenotype [51].” (lines 474).

  1. Engaging the contents of these novel immune phenotypes of COVID-19 with long covid as future research directions within the context of recent literatures will further strength the depth of the manuscript and make it more interesting for the audience to read. One example is that there has been recent literatures (e.g. PMID: 34489601) reporting that monocytes bifurcated into two subpopulations pro-inflammatory and anti-inflammatory subpopulations and they found the quality and quantity of proinflammatory subsets is increased with covid-19 severity. Will any of these monocytes be contributing to the long covid differences? This can be discussed to further elevate the depth of this paper. Similarly, there has already been recent literatures on the novel “adaptive NK phenotype” (PMID: 36066491) and on NK cell with unique early IFNa signatures (PMID: 34592166). These two NK phenotypes are both closely associated with covid-19 severity. It will be one of the important future directions to further evaluate these NK phenotypes in the future long cohorts to study these novel phenotypes’ association with protections.

AUTHORS’ RESPONSE: Thank you for directing us to this exciting field. As this would drive us too far away from our main aim with the paper, we only added this in the discussion by writing “This needs to be addressed on basis of theories and evidence, including psychological and behavioral approaches (e.g., [49]), immune phenotypes [50] and adaptive-like NK cell phenotype [51].” (lines 490).

  1. Accessibility of de-identified data. The data collected from this paper can be informative to the field of long covid research. It will be important for the authors to provide their de-identified data (e.g. symptoms status, vaccination timing, age, gender, etc. ) as supplementary tables. These data is helpful for the reviewers to evaluate their work, and this is also important later for the public to utilize these data to investigate their own hypothesis of interest since some of their more complex hypotheses could be beyond the current manuscript’ scope.

AUTHORS’ RESPONSE: We appreciate this suggestion and have accordingly replaced the Table S1 with such an information (see appendix).

Round 2

Reviewer 2 Report

Accept

Author Response

Thank you very much, we appreciate this option and have asked a native speaker to carefully and extensively clean the text (see in the manuscript with track changes highlighted).

Reviewer 3 Report

This revised manuscript is significantly improved after accounting for the suggestions from this reviewer. However, there are some references seem to be miss-cited and thus needed to be corrected. Also, some other key aspect of long-covid is also missed to analyze/discuss. Thus, this reviewer will suggest another (minor) revision to correct those errors and missing points.

1.       There has been literature (such as PMID: 35216672) studying the connections between levels SARS-CoV2 antibody and autoantibodies and their associations with long covid. The antibody levels of the patients will surely change after vaccination. Will the authors be able to obtain the antibody titer information and integrate that with the long-covid analysis? It is possible that the antibody level induced by vaccination is different than the one induced by real virus infection as studied in PMID: 35216672, and it will be interesting to compare. The reviewers also understand it may not be possible to obtain those antibody titer data. If that’s the case, at least this important connections between antibody levels and long covid should be discussed within the context of the relevant references.

2.       In line 459-462, the authors correctly mentioned “This needs to be addressed on basis of theories and evidence, including psychological and behavioral approaches (e.g., [49]), immune phenotypes [50] and adaptive-like NK cell phenotype [51].”  With their reference 50 and 51. However, their reference 50 is not related to immune phenotype, a correct reference should be PMID: 34489601. Similarly, their reference 51 is also not related to adaptive-like NK cell. Correct references need to be PMID: 36066491 and 34592166. The reviewer suggests the authors proofread and carefully check their references.

Author Response

Reviewer 3

This revised manuscript is significantly improved after accounting for the suggestions from this reviewer. However, there are some references seem to be miss-cited and thus needed to be corrected. Also, some other key aspect of long-covid is also missed to analyze/discuss. Thus, this reviewer will suggest another (minor) revision to correct those errors and missing points.

AUTHORS’ RESPONSE: Thank you very much, we appreciate your careful examination.

  1. There has been literature (such as PMID: 35216672) studying the connections between levels SARS-CoV2 antibody and autoantibodies and their associations with long covid. The antibody levels of the patients will surely change after vaccination. Will the authors be able to obtain the antibody titer information and integrate that with the long-covid analysis? It is possible that the antibody level induced by vaccination is different than the one induced by real virus infection as studied in PMID: 35216672, and it will be interesting to compare. The reviewers also understand it may not be possible to obtain those antibody titer data. If that’s the case, at least this important connections between antibody levels and long covid should be discussed within the context of the relevant references.

AUTHORS’ RESPONSE: Thank you very much, added the following section accordingly:

“Moreover, previous studies (such as [51,53]) investigated the connections between levels of SARS-CoV-2 antibody and autoantibodies as well as post-/long-COVID. Findings (e.g., [26,52]) demonstrate that the antibody levels of the patients change after vaccination. While in the current study, the antibody titer information was not assessed and, accordingly, cannot be analyzed. Future research should investigate whether the antibody level induced by vaccination differs from the one due to the real virus infection to validate earlier research (e.g., [51,53]) and fill this gap with the current study, as antibody titer data was unavailable.” (lines 446ff in the manuscript without track changes).

  1. In line 459-462, the authors correctly mentioned “This needs to be addressed on basis of theories and evidence, including psychological and behavioral approaches (e.g., [49]), immune phenotypes [50] and adaptive-like NK cell phenotype [51].” With their reference 50 and 51. However, their reference 50 is not related to immune phenotype, a correct reference should be PMID: 34489601. Similarly, their reference 51 is also not related to adaptive-like NK cell. Correct references need to be PMID: 36066491 and 34592166. The reviewer suggests the authors proofread and carefully check their references.

AUTHORS’ RESPONSE: Thank you very much, corrected the following section accordingly “This needs to be addressed based on theories and evidence, including psychological and behavioral approaches (e.g., [49]), immune phenotypes [54], and adaptive-like NK cell phenotype [55,56].“ (line 463ff in the manuscript without track changes). Moreover, we checked and corrected the whole manuscript, too (see in the manuscript with track changes highlighted).